# Crystal structure of Zika virus NS5 RNA-dependent RNA polymerase

Andre S. Godoy[1], Gustavo M.A. Lima[1], Ketllyn I.Z. Oliveira[1], Naiara U. Torres[1,2], Fernando V. Maluf[1,2], Rafael V.C. Guido[1] & Glaucius Oliva[1]

The current Zika virus (ZIKV) outbreak became a global health threat of complex epidemiology and devastating neurological impacts, therefore requiring urgent efforts towards the development of novel efficacious and safe antiviral drugs. Due to its central role in RNA viral replication, the non-structural protein 5 (NS5) RNA-dependent RNA-polymerase (RdRp) is a prime target for drug discovery. Here we describe the crystal structure of the recombinant ZIKV NS5 RdRp domain at 1.9 Å resolution as a platform for structure-based drug design strategy. The overall structure is similar to other flaviviral homologues. However, the priming loop target site, which is suitable for non-nucleoside polymerase inhibitor design, shows significant differences in comparison with the dengue virus structures, including a tighter pocket and a modified local charge distribution.

[1] Institute of Physics of São Carlos, University of São Paulo, Av. Joao Dagnone, 1100—Jardim Santa Angelina, São Carlos 13563-120, Brazil. [2] Cellco Biotec, R. Alberto Lanzoni, 993—Parque Santa Felicia, São Carlos 13562-390, Brazil. Correspondence and requests for materials should be addressed to G.O. (email: oliva@ifsc.usp.br).

Though initially described as a mild version of dengue fever, the Zika virus (ZIKV) outbreak in the Americas unexpectedly revealed major neurological impacts as fetal microcephaly or other congenital brain injury when women are infected during pregnancy and Guillain-Barré syndrome in adults[1,2]. Differently from other flaviviruses, ZIKV is transmitted both by the insect vector and sexual contact, and the virus has been found for months in semen of infected patients[3]. In parallel to the development of a vaccine, there is an urgent need for novel safe and effective antiviral drugs both for treatment and prophylaxis of ZIKV infection.

The flaviviral non-structural protein 5 RNA-dependent RNA-polymerase (NS5 RdRp) has a central role in virus genome replication and is absent in the mammalian hosts, being extensively targeted for drug discovery and development[4]. Although nucleoside polymerase inhibitors (NPIs) have achieved clinical success in the case of Hepatitis C virus infections (for example, sofosbovir), they depend on activation by host kinases and are potentially subjected to toxicity problems[5]. Therefore, non-NPIs have been actively sought as inhibitors of flaviviral NS5 RdRp, particularly targeting the so-called priming loop, that regulates RNA-template binding and polymerization[6]. Recently, several groups reported the discovery of novel RdRp inhibitors with pan-serotype activity against dengue viruses (DENV). In these cases, the use of X-ray crystallographic structures was fundamental to develop optimized lead candidates[7,8]. Herein we describe the crystal structure of the ZIKV NS5 RdRp domain and compare it with the homologous dengue virus proteins from different serotypes to identify suitable target sites for anti-ZIKV drug discovery and elucidate their structural drug-binding features.

## Results

**Overall structure of ZIKV NS5 RdRp.** We recombinantly expressed and purified ZIKV NS5 RdRp (residues 306–903) of the MR/766 strain. Purified protein was crystallized and its three-dimensional structure was determined by molecular replacement. The structure of the NS5 RdRp was refined to 1.9 Å resolution, with final $R_{work}$ and $R_{free}$ of 17 and 20%, respectively (Table 1). Similar to other flaviviruses, ZIKV NS5 RdRp adopts a 'right-hand'-shaped structure, with the three main domains named as fingers (residues 321–488 and 542–608), palm (residues 489–541 and 609–714) and the thumb (residues 715–903). The active site of the ZIKV NS5 RdRp is located above the palm and is surrounded by loops protruding from both thumb and palm domains, likely to be involved in the stabilization of the RNA molecules during the extension (Fig. 1a). The first 15 N-terminal and last 16 C-terminal residues are not visible in the electron density. Due to disorder, parts of loops 1 (residues 340–363), 3 (residues 408–417), 4 (residues 454–471), 5 (residues 533–542) and 6 (residues 576–606) are also missing in the model. Similar to DENV, the ZIKV NS5 RdRp binds to two $Zn^{+2}$ ions, one at the finger (G439, H443, C448 and C451) and the other at the thumb (H714, C730, C849) domain (Fig. 2). Crystals were obtained in large numbers and diffracted to high resolution. The active site is well exposed to the crystal solvent channels and preliminary soaking experiments did not affect the crystal packing or the diffraction limit, which altogether represents a solid platform for subsequent structure-based drug design strategies.

**Active site and priming loop.** The mechanism of action of RNA polymerases requires two aspartic acids associated with the binding and positioning of two metal ions that catalyse the nucleotidyl transfer[9]. In ZIKV NS5 RdRp, they correspond to aspartates 535 (disordered in the structure) and 665, respectively.

**Table 1 | Data collection and refinement statistics.**

|  | ZIKV NS5 RdRp |
|---|---|
| *Data collection* | |
| Space group | $P4_32_12$ |
| Cell dimensions | |
| $a, b, c$ (Å) | 78.9, 78.9, 210.02 |
| $\alpha, \beta, \gamma$ (°) | 90.0, 90.0, 90.0 |
| Resolution (Å) | 29.66–1.9 (1.95–1.9) |
| $R_{p.i.m.}$ | 0.074 (0.85) |
| $I/\sigma I$ | 9.9 (1.8) |
| Completeness (%) | 99.7 (96.1) |
| Redundancy | 24.3 (11.1) |
| $CC^{1/2}$ | 0.99 (0.72) |
| | |
| *Refinement* | |
| Resolution (Å) | 29.66–1.9 |
| No. of reflections | 53,499 |
| $R_{work}/R_{free}$ | 0.17/0.20 |
| No. of atoms (non-H) | |
| Protein | 3,878 |
| Ligands | 7 |
| Water | 663 |
| *B-factors* | |
| Protein | 31.7 |
| Ligands | 73 |
| Water | 44 |
| r.m.s.d. | |
| Bond lengths (Å) | 0.004 |
| Bond angles (°) | 0.58 |

As observed for other RdRp from flaviviruses, the ZIKV NS5 RdRp active site is located at the intersections of two tunnels (Fig. 1b). It was proposed that the first tunnel, formed by the interfaces of the fingers and the thumb domain, is responsible for coordinating the single-strand RNA, while the second tunnel coordinates the nascent double-strand RNA[9]. The priming loop, responsible for the allosteric positioning of the 3′ terminus of the RNA template at the active site is also present in ZIKV NS5 RdRp (residues V785–D810, Fig. 1a and Supplementary Fig. 1).

## Discussion

The construct used in the present work was previously described in the paper of Malet *et al.*[10], in which they crystallized two forms of the West Nile Virus (WNV) NS5 RdRp named POL1 (res 273–905) and POL2 (res 317–905). Their POL1 structure (PDB 2HFZ) only achieved a moderated resolution (3.0 Å), whereas POL2 (PDB 2HCN) was determined at 2.3 Å resolution. By comparing the structures of POL1 and POL2, Malet *et al.*[10] concluded that there are no differences in the orientation of the residues within the active site between both constructs, whereas POL2 is a more suitable platform for rational design of non-NPIs. Indeed, ZIKV NS5 RdRp is a high-resolution model when compared with the structure of full-length ZIKV NS5 (PDB 5TFR at 3.05 Å resolution, not yet published).

Overall, the ZIKV NS5 RdRp structure is highly similar to other known flaviviral homologues (Fig. 3). The Japanese Encephalitis (JEV) and the WNV are the closest homologues, both with 70% sequence identity with ZIKV NS5. Notwithstanding, the overall structures of JEV and WNV RdRp are very similar to the one from ZIKV (r.m.s.d. of 1.1 and 1.2 Å, respectively)[10,11]. The ZIKV thumb, palm and fingers domains are also well superimposed on the equivalent domains of DENV2 and DENV3 (r.m.s.d. of 1.17 Å and 1.19 Å, respectively)[6,12]. Nevertheless, a close inspection at the priming loop revealed structural elements that differ from DENV homologues and

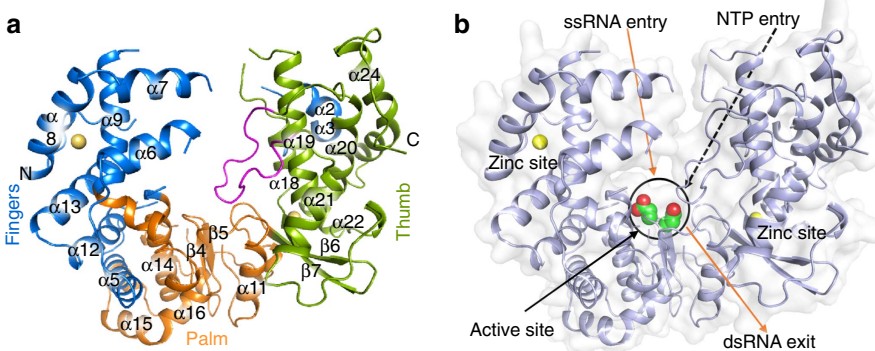

**Figure 1 | Crystal structure of Zika virus NS5 RdRp.** (**a**) ZIKV NS5 RdRp structure with fingers, palm and thumb domains coloured in blue, orange and green, respectively. The priming loop is depicted in pink. (**b**) Surface view of the ZIKV NS5 RdRp, with orange arrows pointing the entry of the single-strand RNA template, and the exit region of the double-strand RNA. Catalytic aspartates are depicted as green/red spheres. Black dashed arrow shows the entry path of NTPs, while black arrow point to the position of the active site. In both figures, zinc atoms are depicted in yellow.

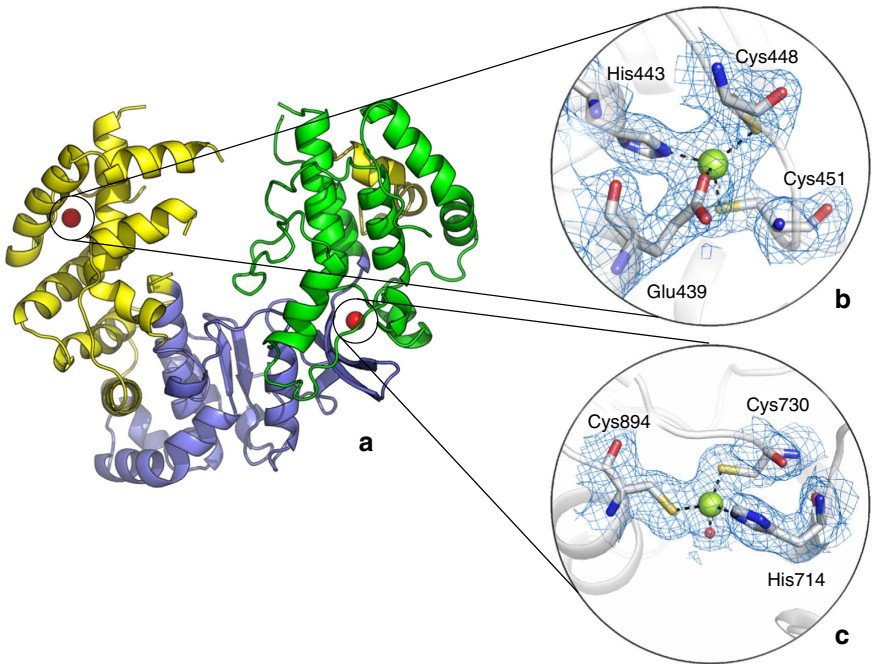

**Figure 2 | Overall and detailed view of the ZIKV NS5 RdRp.** (**a**) General view of the ZIKV NS5 RdRp coloured by thumb (green), fingers (yellow) and palm (purple) domains. Zinc atoms are coloured as red spheres. (**b**) Details of the fingers domain zinc-binding site shows main coordinating amino acids (E439, H443, C448 and C451) depicted as sticks. (**c**) Details of the thumb domain zinc-binding site shows main coordinating amino acids (H714, C730 and C894) depicted as sticks. Blue mesh represents the 2Fo-Fc with 1.0 sigma.

might impact on inhibitor discovery. The sequence identities of ZIKV with DENV2 and DENV3 priming loops are 76 and 81%, respectively. The main differences lie on residues in the beginning of the sequence (ZIKV: V787, D788, G793; DENV2: S785, H786, S791; DENV3: H786, S791, respectively), which are solvent-exposed, and at the end of the loop (ZIKV: G801, G803; DENV2: A799, H801; DENV3: A799, H801, respectively). These mutations determine structural differences in the conformation of ZIKV NS5 RdRp compared with DENV2 and DENV3. The priming loop of the ZIKV adopts a closed conformation when compared with the equivalent region in DENV2 (ligand bound) and DENV3 (ligand free and ligand bound), providing a smaller binding pocket (Fig. 4a). ZIKV RdRp priming loop has three glycine residues (G793, G801 and G803, respectively), which are replaced with bulkier amino-acid residues in DENV2 and DENV3 (S791, A799 and H801, respectively). These substitutions may provide additional flexibility to the ZIKV priming loop,

thereby allowing it to adopt a closed conformation, similar to the ones adopted by JEV and WNV (Fig. 5)[6,10,11]. The closed conformation renders the access to a deeper narrow cavity close to E804 (Q802 in DENV3 and E802 in DENV2) (Fig. 4b). Moreover, the neutral side chains of the G801 and G803 residues impact on the overall charge distribution in the ZIKV priming loop binding site. While electrostatic surface of DENV2 and DENV3 are positively charged, the ZIKV priming loop binding site presents more neutral surface charge (Fig. 4b). Molecular interactions with amino acids of this binding site are important to generate both potent and pan-serotype active NS5 RdRp inhibitors[6,12,13]. Therefore, these findings indicate some structural determinants that can be explored for the discovery and development of new antiviral against ZIKV NS5 RdRp.

In conclusion, we report a high-resolution crystal structure of the ZIKV NS5 RdRp domain. Despite the structural similarities with DENV, the ZIKV NS5 RdRp exhibits significant differences

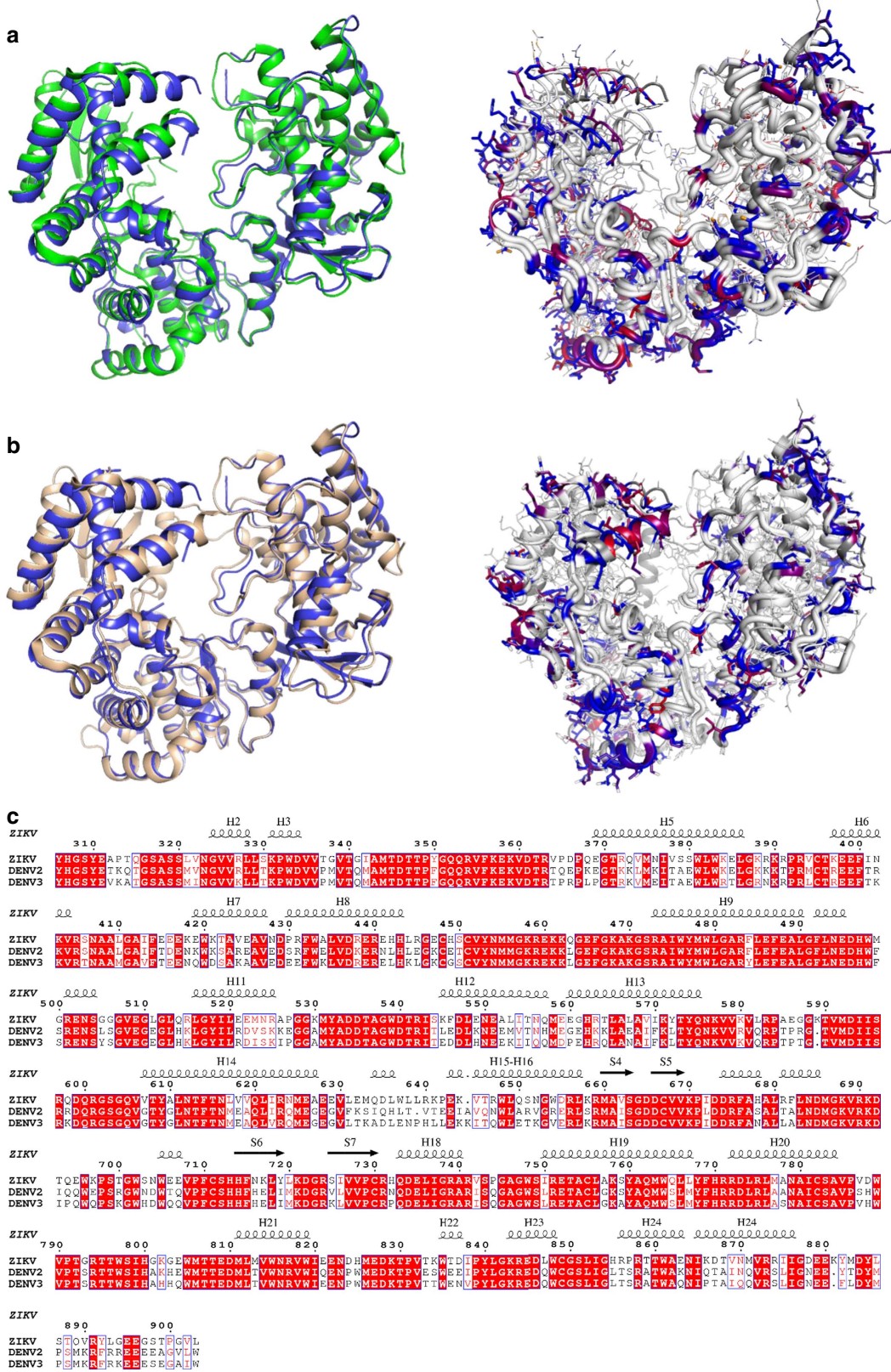

**Figure 3 | Primary and tertiary structural comparison of Zika and dengue virus NS5 RdRp.** (**a**) Superposition (left) and coloured differences (right) of the NS5 RdRp structures of DENV2 (green) and ZIKV (blue). (**b**) Superposition (left) and coloured differences (right) of the NS5 RdRp structures of DENV3 (wheat) and ZIKV (blue). In the right panel, mutations are coloured by percentage of difference in a blue–red scale, based on a BLOSUM90 substitution matrix. (**c**) Alignment of the NS5 RdRp sequences of ZIKV, DENV2 and DENV3, with secondary structure of ZIKV depicted at the top. Red boxes show conserved residues, while blue boxes show conserved substitutions. Helix were named with H, while strands with S.

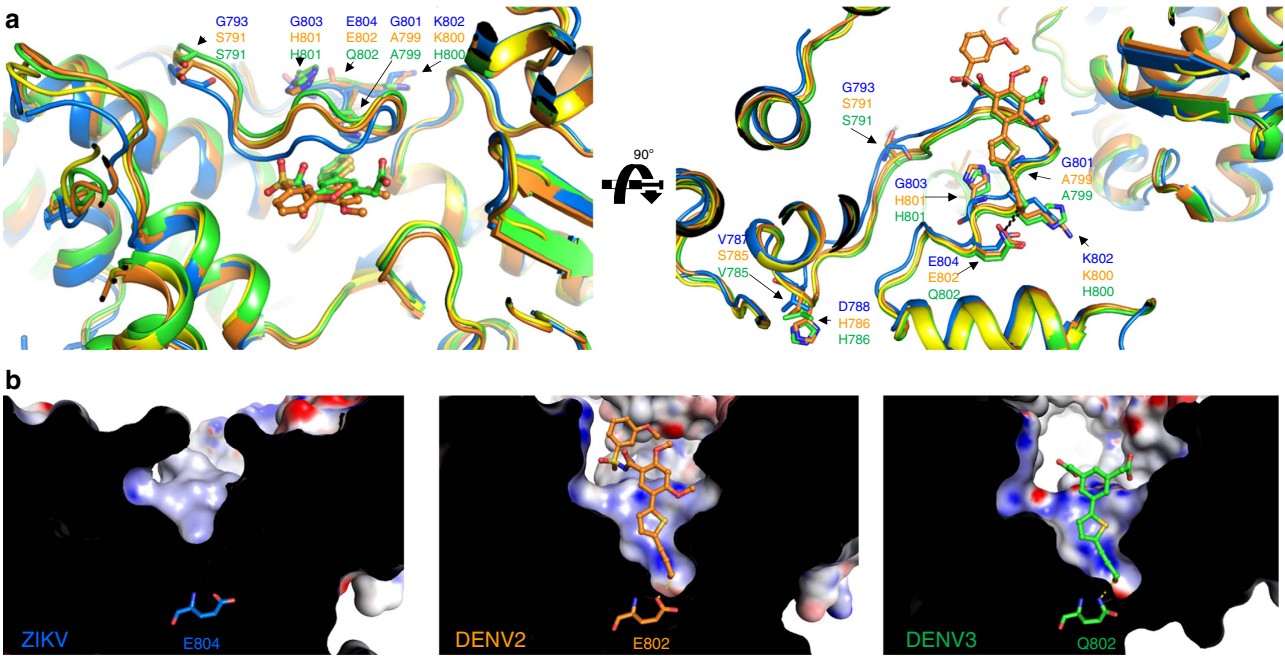

**Figure 4 | Structural comparison of Zika and dengue virus NS5 RdRps.** (**a**) Orthogonal view of superimposed priming loops in NS5 RdRp domain from ZIKV (5U04, blue), DENV2 in complex with 5-[5-(3-hydroxyprop-1-yn-1-yl)thiophen-2-yl]-2,4-dimethoxy-N-[(3-methoxyphenyl)sulfonyl]benzamide (5K5M, orange) and DENV3 in complex with 2,2'-(5-(5-(3-hydroxyprop-1-yn-1-yl)thiophen- 2-yl)-1,3-phenylene)diacetic acid (5HMY, green) and DENV3 apo (2J7U, yellow). (**b**) Close-up views of the priming loop binding sites of ZIKV, DENV2 and DENV3 showing the surface charge distributions (same view as in (**a**) right panel). Positive and negative electrostatic potential are indicated in blue ($+50\,kTe^{-1}$) and red ($-50\,kTe^{-1}$), respectively.

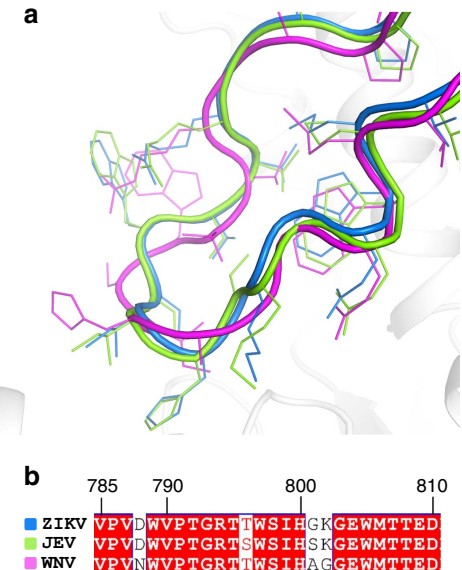

**Figure 5 | Tertiary structural comparison of flaviviral priming loops.** (**a**) Superposition of ZIKV (blue), JEV (green) and WNV (magenta) priming loops. (**b**) Alignment of the NS5 RdRp sequences of ZIKV, JEV and WNV priming loops. Red boxes show conserved residues.

in the priming loop binding site that may affect ligand design. Therefore, our findings can pave the way to the discovery and development of broad-spectrum antiviral drug candidates against *Flaviviridae*, including ZIKV.

## Methods

**Recombinant source of ZIKV-NS5RdRp protein.** A synthetic DNA fragment of the ZIKV-NS5RdRp (GenBank: EU303241.1) gene was codon optimized and synthesized by Genscript (Supplementary Note 1). To facilitate protein

crystallization, the N-terminal portion (residues 272–305) was removed from the ZIKV NS5 RdRp[10]. In WNV NS5 RdRp, this construct (POL1) generated better diffracting crystals than the full-length RdRp domain. The fragment was amplified and cloned using the primers NS5RdRp_Fw (5'-CAGGGCGCCATGTACCATGG GAGCTACGAAGC-3') and NS5RdRp_Rv (5'-GACCCGACGCGGTTACAACAC TCCGGGTGTGG-3'). The fragment was cloned into the expression vector pETTrx-1a/LIC by ligation-independent cloning[14]. For that, gene product and the linearized vector were treated with 0.33 units of T4 DNA Polymerase (Fermentas) in the presence of $1\times$ T4 Polymerase Buffer, $4.0\,mmol\,l^{-1}$ dithiothreitol and $2.5\,mmol\,l^{-1}$ dATP (for the gene) or $2.5\,mmol\,l^{-1}$ dTTP (for the vector). Samples were incubated for 30 min at 22 °C and heat inactivated at 75 °C for 20 min. A volume of $1.0\,\mu l$ of T4 polymerase-treated vector was mixed to $3.0\,\mu l$ of T4 polymerase-treated PCR fragment and incubated at 25 °C for 30 min, which was followed by chemical transformation in DH10B *Escherichia coli* and BL21(DE3) cells. Positive colonies were selected by colony PCR. Rosetta2 *(DE3) pLysS E. coli* (Novagen) carrying ZIKV-NS5RdRp-pETTrx plasmid were cultured at 37 °C, with shaking at 200 r.p.m., in LB medium supplemented with $50\,\mu g\,ml^{-1}$ kanamycin and $34\,\mu g\,ml^{-1}$ chloramphenicol until $DO_{600}$ of 0.5 was reached. Expression was induced with $0.5\,mmol\,l^{-1}$ isopropylthiogalactoside, and the temperature was subsequently reduced to 18 °C for 16 h. Cells were harvested by centrifugation ($3,500g$ for 30 min at 4 °C). Cells pellets were resuspended in lysis buffer ($50\,mmol\,l^{-1}$ Tris pH 7.5, $500\,mmol\,l^{-1}$ NaCl, 10% glycerol, $5.0\,mmol\,l^{-1}$ $MgSO_4$) containing $1.0\,mmol\,l^{-1}$ dithiothreitol, $1.0\,mmol\,l^{-1}$ phenylmethyl sulfonyl fluoride, $200\,\mu g\,ml^{-1}$ lysozyme and $3.0\,U\,ml^{-1}$ nuclease from *Serratia marcescens* and lysed by sonication on ice. Insoluble debris was separated by centrifugation ($20,000g$, 30 min, 4 °C) and the soluble fraction was loaded onto a HisTrap HP 5.0 ml (GE Healthcare). The His-tagged protein was eluted with a $0–300\,mmol\,l^{-1}$ imidazole gradient in the same buffer and then buffer exchanged ($50\,mmol\,l^{-1}$ Tris pH 7.5, $150\,mmol\,l^{-1}$ NaCl) by desalting with Superdex G-25 Fine (GE Healthcare). The $His_6$-Trx tag was cleaved with TEV protease (1.0 mg per 20 mg of ZIKV-NS5RdRp, 16 h, 4 °C), and the protein mixture was reloaded on the HisTrap column to remove the cleaved $His_6$-Trx tag and any uncleaved protein. The cleaved protein was further purified by size-exclusion chromatography on a HiLoad 16/60 Superdex 75 column (GE Healthcare) pre-equilibrated in buffer $20\,mmol\,l^{-1}$ HEPES, pH 7.5, $200\,mmol\,l^{-1}$ NaCl and 5% glycerol. Protein concentration was determined spectrophotometrically using a theoretical extinction coefficient of $163,330\,mol^{-1}\,cm^{-1}$ at 280 nm calculated using ProtParam[15]. Protein purity was confirmed by SDS–PAGE and concentrated to $6.0\,mg\,ml^{-1}$.

**Crystallization and data collection.** Crystallization screening was performed with the sitting drop vapour diffusion method in 96-well plates using a Phoenix Liquid Handling System—Gryphon LCP (Art Robbins Instruments) and commercially available screens. The trials were set with 200 nl of protein solution and an equivalent volume of reservoir, equilibrated against 80 µl reservoir at 18 °C.

Crystals were observed after 2 days in condition C6 of Morpheus Screen (Molecular Dimensions) containing 0.3 mol l$^{-1}$ sodium nitrate, 0.3 mol l$^{-1}$ Sodium phosphate dibasic, 0.3 mol l$^{-1}$ ammonium sulfate, Morpheus Buffer System 2 pH 7.5 (Sodium HEPES; MOPS) and 40% v/v ethylene glycol; 20% w/v PEG 8000. Single crystals were cryo-cooled in liquid nitrogen and X-ray diffraction data were collect at beam line MX2 at the Brazilian Synchrotron Light Laboratory using a wavelength of 1.4586 Å. Data were indexed, integrated and scaled in XDS[16] and AIMLESS[17]. The crystals are tetragonal with space group $P4_32_12$ and unit cell dimensions of $a = b = 78.9$ Å, $c = 210.02$ Å, $\alpha = \beta = \gamma = 90°$. The molecular weight of a subunit is 68.9 kDa, and the asymmetric unit consist of one protein molecule with the solvent content of approximately 49%. Data collection statistics are given in Table 1.

**Structure solution and refinement.** The ZIKV-NS5RdRp was solved by molecular replacement by using a monomer from WNV RdRp structure (sequence identity of 72%, PDB code 2HCN[10]) as the search model and carried out in Phaser[18]. The structure was initially rebuilt using the AutoBuild wizard of PHENIX[19–21]. The model was then improved by iterative manual building into $2F_o - F_c$ and $F_o - F_c$ electron density maps using Coot[22]. Five percent of reflections were set aside from refinement for calculation of $R_{free}$ and reciprocal-space restrained refinement was carried out with phenix.refine[21], where torsion/libration/screw motion restraints (TLS) restraints were applied throughout the final steps of the refinement. Data processing and refinement statistics are summarized in Table 1. Model quality was checked using MolProbity[23]. The final model was refined to 1.9 Å and has good stereochemistry, with 98.5% of the residues in the most favoured regions of the Ramachandran plot. All molecular figures were prepared in PyMOL (Schrödinger, LLC). Zinc sites had poor anomalous signal and were validated with CheckMyMetal[24].

**Data availability.** Coordinates and structure factor have been deposited in the Protein Data Bank under accession codes 5U04. The PDB accession codes ZIKV (5TFR), DENV2 (5K5M), DENV3 (5HMY and 2J7U), ZIKV (5TIT), WNV (2HCN), JEV (4MTP) and GenBank entry EU303241.1 were used in this study. All other data are available from the corresponding author upon reasonable request.

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

## Acknowledgements

We thank H.M. Pereira for technical assistance with the crystallization robot facility (EMU/FAPESP grant 2015/16811-3) and J.R.C. Muniz for data collection support at the synchrotron beamline MX2 (LNLS, Campinas). Funding from the São Paulo Research Foundation—FAPESP (CEPID grant 2013/07600-3, PIPE grant 2014/50381-3, and fellowships 2016/19712-9, 2016/17153-2) is gratefully acknowledged.

## Author contributions

A.S.G., G.O. and R.V.C.G. conceived, coordinated and wrote the manuscript. A.S.G. solved the X-ray structure. A.S.G., G.M.A.L. and K.I.Z.O. performed purification and crystallization of the macromolecule. F.V.M. and N.U.T. prepared the soluble construct of the macromolecule.

## Additional information

**Competing financial interests:** The authors declare no competing financial interests.

**Publisher's note**: 

