## [Peer review file · Nature Communications]

Reviewers' comments:

Reviewer #1 (Remarks to the Author):

General comments: Godoy et al. solved the crystal structure of Zika virus RdRp domain. They compared this structure with DENV2 and DENV3 RdRp structures and determined that there is high homology except in the priming loop region. The Zika RdRp structure is of relatively good quality (refined to 2.3 angstrom), however, the number of aa residues analysed for (Ramachandran outliers, rotameric score, model fitting) are no more than 75% of the total aa, which skews the statistical analyses. There are multiple regions that are disordered (pg 3 lines 57-59). This protein is missing aa residues 273-304 (part of finger subdomain) which may influence the conformation of the priming loop region. Given the higher similarity of Zika RdRp with JEV and WNV RdRp, the authors should also compare the conformation of the Zika priming loop with these two Flaviviruses. A difference in the relative positions of WNV and DENV3 priming loops was previously observed (Malet et al. 2008. Antiviral Res. 80: 23-35; Lim et al., 2016, PLoS Pathogens, 12(8):e1005737). Section on pg 4, lines 83-87 should thus be amended.

Specific comments:

1. Pg 2, line 40: the wrong reference is cited, the correct ref is Lim et al 2016, PLoS Pathogens, 12(8):e1005737.
2. Pg 3, line 43: the wrong reference is cited, the correct refs are 6, 8.
3. Pg 3, zika RdRp structure: is part of the C or N terminal also disordered? If so, these should be stated.
4. Pg 4, lines 77 and pg 5, line 98: the wrong reference (10) is cited, the correct ref is 11 and Lim et al 2016, PLoS Pathogens, 12(8):e1005737.
5. Supplementary material, structure solution and refinement: which Flavivirus RdRp was used in the search model, this should be stated.

Reviewer #2 (Remarks to the Author):

The authors report the crystal structure of the NS5 RNA-dependent RNA-polymerase domain (not full-length) of ZIKV at 2.3 Å.

Collection and refinement statistics are satisfactory. However, a significant number of amino-acids are missing (88 residues, in 5 loops).

The authors compare their structure with known Flavivirus structure, and in particular with DENV NS5, and conclude that the overall structure is similar.

Despite the structural similarities with DENV, the ZIKV NS5 RdRp the authors mention differences in the priming loop binding. They speculate that these differences... "may affect ligand design".

The interest of the work stems from the fact that ZIKV is a significant pathogen receiving a lot of public attention.

However, the work does not add any significant knowledge to what is known in the Flavivirus NS5 field. The structure cannot be qualified as « high-resolution » structure, as the authors do in the conclusion. No effort is made to identify the putative Zn ion using anomalous diffraction Existing crystal structures of related DENV and JEV full-length NS5 are in fact more informative. Published work is far more discussed and analyzed than what is presented here.

The authors do not indicate if the crystallized protein has enzymic activity, which is of key importance if this work is meant to help drug design..

Reviewers' comments:

Reviewer #1 (Remarks to the Author):

General comments: Godoy et al. solved the crystal structure of Zika virus RdRp domain. They compared this structure with DENV2 and DENV3 RdRp structures and determined that there is high homology except in the priming loop region.

- The Zika RdRp structure is of relatively good quality (refined to 2.3 angstrom), however, the number of aa residues analysed for (Ramachandran outliers, rotameric score, model fitting) are no more than 75% of the total aa, which skews the statistical analyses. There are multiple regions that are disordered (pg 3 lines 57-59).

A: The loops mentioned are also disordered in other examples of NS5 RdRp of all other Flavivirus (i.e PDB 2HCN, 2HCS, 2HFZ, 5HMX, 5HMW, etc). The high mobility and consequently the high B-factors for these loops is probably required for substrate recognition, binding and accommodation at the active site, and is a standard for x-ray structures of the RdRp domain. Notwithstanding, some loops (i.e 340-363) are expected to form tertiary structure arrangements with the Methyltransferase domain of the Full length NS5. Therefore, these loops are expected to be disordered in our RdRp domain structure, now solved at 1.9A resolution, but this should not affect the drug design efforts aimed at the active site components.

Fig.1 . B-factor putty of RdRp domains of known flavivirus NS5 (including ZIKV). Circled show common missing loops.

This protein is missing aa residues 273-304 (part of finger subdomain) which may influence the conformation of the priming loop region.

A: The X-ray structures of JEV RdRp (PDB 4MTP – construct 273-905) and WNV (PDB 2HCN – construct 317-905) have similar positions for the priming loop with ZIKV. This relative region of the subdomain is present in the JEV x-ray structure, and does not seem to influence the priming loop position (Fig. 2). We present a

higher resolution structure, similar to the shorter construction described for WNV (2007, J.Biol.Chem. 282: 10678-10689).

Fig. 2. ZIKV (green) and JEV (blue) x-ray structures. Priming loop is circled in red

Given the higher similarity of Zika RdRp with JEV and WNV RdRp, the authors should also compare the conformation of the Zika priming loop with these two Flaviviruses. A difference in the relative positions of WNV and DENV3 priming loops was previously observed (Malet et al. 2008. Antiviral Res. 80: 23-35; Lim et al., 2016, PLoS Pathogens, 12(8):e1005737). Section on pg 4, lines 83-87 should thus be amended.

Specific comments:

1. Pg 2, line 40: the wrong reference is cited, the correct ref is Lim et al 2016, PLoS Pathogens, 12(8):e1005737.

A: fixed according reviewer suggestion

2. Pg 3, line 43: the wrong reference is cited, the correct refs are 6, 8.

A: fixed according reviewer suggestion

3. Pg 3, zika RdRp structure: is part of the C or N terminal also disordered? If so, these should be stated.

A: fixed according reviewer suggestion

4. Pg 4, lines 77 and pg 5, line 98: the wrong reference (10) is cited, the correct ref is 11 and Lim et al 2016, PLoS Pathogens, 12(8):e1005737.

A: fixed according reviewer suggestion

5. Supplementary material, structure solution and refinement: which Flavivirus RdRp was used in the search model, this should be stated.

A: fixed according reviewer suggestion

Reviewer #2 (Remarks to the Author):

The authors report the crystal structure of the NS5 RNA-dependent RNA-polymerase domain (not full-length) of ZIKV at 2.3 Å.

Collection and refinement statistics are satisfactory. However, a significant number of amino-acids are missing (88 residues, in 5 loops).

A: The loops mentioned are also disordered in other examples of NS5 RdRp of all other Flavivirus (i.e PDB 2HCN, 2HCS, 2HFZ, 5HMX, 5HMW, etc). The high mobility and consequently the high B-factors for these loops is probably required for substrate recognition, binding and accommodation at the active site, and is a standard for x-ray structures of the RdRp domain. Notwithstanding, some loops (i.e 340-363) are expected to form tertiary structure arrangements with the Methyltransferase domain of the Full length NS5. Therefore, these loops are expected to be disordered in our RdRp domain structure, now solved at 1.9Å resolution, but this should not affect the drug design efforts aimed at the active site components

Fig.1 . B-factor putty of RdRp domains of known flavivirus NS5 (including ZIKV). Circled show common missing loops.

The authors compare their structure with known Flavivirus structure, and in particular with DENV NS5, and conclude that the overall structure is similar.

Despite the structural similarities with DENV, the ZIKV NS5 RdRp the authors mention differences in the priming loop binding. They speculate that these differences... "may

affect ligand design". The interest of the work stems from the fact that ZIKV is a significant pathogen receiving a lot of public attention. However, the work does not add any significant knowledge to what is known in the Flavivirus NS5 field. The structure cannot be qualified as « high-resolution » structure, as the authors do in the conclusion.

A: We obtained a significantly better dataset of our crystal (1.9 Å resolution, R_{work}/R_{free} of 18/20) and decided to update the manuscript with this new data. There are no differences between new and old data regarding disordered loops, or any other residue (RMSD of 0.15).

No effort is made to identify the putative Zn ion using anomalous diffraction

A: The re-processed data shows a very weak anomalous signal, with only 1.0 d''/σ at 8.0 Å resolution (Figure below)

This is probably due to the wavelength used at the data collection (1.47 Å), where the anomalous scattering of Zn is only $\sim 2e$ (f' 0.72, f'' -1.8). For that, it was not possible to solve the structure by the anomalous signal of the Zn. Nevertheless, the calculated anomalous map from the refined molecular replacement model shows clearly the intensity peaks of the Zn at the site positions (figure below).

To validate the Zn atoms, we check the stereochemistry of the Zn bonds using CheckMyMetal (http://csgid.org/csgid/metal_sites/). The analysis suggests good

quality for the coordination and valence of the metal sites, with no alternatives metals for it (figure below).

CheckMyMetal(CMM) Home		Report a Problem										
ID	Res.	Metal	Occupancy	B factor (env.) ¹	Ligands	Valence ²	nVECSUM ³	Geometry ^{1,4}	gRMSD(°) ¹	Vacancy ¹	Bidentate	Alt. metal
B:1	ZN	Zn	1	28.1 (29.6)	O ₁ N ₁ S ₂	1.9	0.025	Tetrahedral	9.6°	0	0	
B:2	ZN	Zn	1	41.8 (46.4)	O ₁ N ₁ S ₂	1.8	0.046	Tetrahedral	8.8°	0	0	

Legend: Not applicable **Outlier** Borderline Acceptable

Generate a model with alt. metal: Select the metal(s) above and

Existing crystal structures of related DENV and JEV full-length NS5 are in fact more informative. Published work is far more discussed and analyzed than what is presented here. The authors do not indicate if the crystallized protein has enzymic activity, which is of key importance if this work is meant to help drug design.

A: 3) The expression vector construction that we adopted was previously described in the paper of Malet et al. (JBC 282, p 0678-89, 2007), in which they crystallized two constructions of the West Nile Virus NS5 RdRp, named POL1 (res 273-905) and POL2 (317-905). Their POL1 structure (PDBcode) only achieved a moderated resolution of 3.0 Å, whereas POL2 (PDBcode) reached 2.3 Å resolution. By comparing the structures of POL1 and POL2, Malet et al. concluded that “there are no differences in the orientation of the residues within the active site between POL2 and POL1”. There are no reasons to believe that the ZIKV equivalent constructions will be any different. As our major goal with this study is to provide a solid platform for drug design against ZIKV NS5 RdRp, we decided to express the ZIKV equivalent shorter version of WNV POL2.

As for the enzymatic activity of this shorter construct, the POL2 of WNV NS5 RdRp is “unable to incorporate [3H]GMP using the homopolymeric RNA template poly(rC)”, but rendered more suitable crystals of the enzyme. Indeed, the good quality model (1.9Å resolution) presented in our manuscript is a key component for the rational design of non-nucleoside polymerase inhibitors. The other ZIKV NS5 full length structure deposited at PDB (5TFR, not published yet) shows only the modest resolution of 3.05Å, far from the ideal for computational screening of enzyme inhibitors and crystallographic fragment-based studies.

REVIEWERS' COMMENTS:

Reviewer #1 (Remarks to the Author):

The quality of the crystallography work is improved but the overall findings are not sufficiently novel. There is an absence of functional data on the enzyme activities, or inhibition by compounds to provide insights for drug discovery.

Reviewer #2 (Remarks to the Author):

The authors have adequately answered all questions and significantly improved the ms, which is now in a form ready to be published.